# Feedback from HTC Vive Sensors Results in Transient Performance Enhancements on a Juggling Task in Virtual Reality

**DOI:** 10.3390/s21092966

**Published:** 2021-04-23

**Authors:** Filip Borglund, Michael Young, Joakim Eriksson, Anders Rasmussen

**Affiliations:** 1Virtual Reality Laboratory, Design Sciences, Faculty of Engineering, Lund University, 221 00 Lund, Sweden; filip.borglund.5685@student.lu.se (F.B.); michael.young.3236@student.lu.se (M.Y.); joakim.eriksson@design.lth.se (J.E.); 2Department of Neuroscience, Erasmus University Medical Center, 3000 DR Rotterdam, The Netherlands; 3Associative Learning, Department of Experimental Medical Science, Medical Faculty, Lund University, 221 00 Lund, Sweden

**Keywords:** juggling, feedback, HTC Vive, learning, timing, virtual reality

## Abstract

Virtual reality headsets, such as the HTC Vive, can be used to model objects, forces, and interactions between objects with high perceived realism and accuracy. Moreover, they can accurately track movements of the head and the hands. This combination makes it possible to provide subjects with precise quantitative feedback on their performance while they are learning a motor task. Juggling is a challenging motor task that requires precise coordination of both hands. Professional jugglers throw objects so that the arc peaks just above head height, and they time their throws so that the second ball is thrown when the first ball reaches its peak. Here, we examined whether it is possible to learn to juggle in virtual reality and whether the height and the timing of the throws can be improved by providing immediate feedback derived from the motion sensors. Almost all participants became better at juggling in the ~30 min session: the height and timing of their throws improved and they dropped fewer balls. Feedback on height, but not timing, improved performance, albeit only temporarily.

## 1. Introduction

Rapid technological advancements have enabled researchers to obtain rich data about our own physiological state and movements. We can get feedback on activity levels and heart rate from smart phones and watches. Other sensors can give us information about muscle activity or limb movements. Accumulating evidence shows that providing feedback derived from movement sensors can enhance motor learning. For example, feedback from Electromyography (EMG) electrodes can help singers adjust the tone of muscles involved in singing [1]. Feedback can also enhance postural control in elders [2]. Compared to qualitative feedback, providing quantitative feedback enhances learning of the crawl stroke in swimming [3]. Studies also show that feedback can improve the timing of movements. Thus, providing participants with data from periocular EMG data allows them to control the closure of their eyes with impressive timing accuracy [4]. There is even speculation that one could use feedback in an immersive virtual reality (VR) environment to train empathy [5].

Virtual reality (VR) refers to technology that enables human–computer interaction that simulates as many senses as possible. More precisely, it is commonly used to describe interactive computer-generated 3D environments that the user can interact with and that incorporate sensory feedback. Other related technologies include augmented reality (AR), in which digital content is superimposed on a user view of the real world, and mixed reality (MR), in which the real world and digital context co-exist and interact [6]. Recently, VR products utilizing some form of head mounted display (HMD) and positional tracking, such as the Oculus Rift and HTC Vive headsets, have grown in popularity. In this study, we use the HTC Vive, which has an accuracy that is superior to that of other HMDs, making it a promising tool for scientific and clinical purposes [7,8]. 

Juggling can refer to any type of activity that involves object manipulation. However, the word is most commonly used to describe toss juggling, which means throwing and catching objects (usually balls or clubs) in different patterns. Juggling is a mentally challenging task that requires advanced sensorimotor coordination [9]. Practicing juggling leads to changes in cortical white matter architecture lasting at least four weeks, although the changes do not correlate with performance [10]. To succeed, it is important to throw the projectiles in a consistent pattern and to pay close attention to the timing and the height of the throws. Throwing the projectiles with shallow trajectories result in smaller errors, but it also increases the risk of collisions and reduces the amount of time that the juggler has to deal with the objects. A higher trajectory results in more time to correct mistakes but also a greater amplification of errors. The optimal height to throw the objects varies depending the number of objects—more objects necessitates higher throws [11]. While some professional jugglers can juggle using only proprioception and somatosensory input, novice and intermediate jugglers are dependent on their vision and cannot juggle for more than a few seconds if they close their eyes [12]. For this overwhelming majority, it is helpful if the zenith of the objects’ trajectories is within their visual field. Indeed, aspiring jugglers are often advised to look at the highest point of the trajectory and throw the next ball when the previous ball reaches the top [11,13]. The timing and height of throws can be taught by trial-and-error or by having a professional juggler tell the aspiring juggler when a throw was too high/low or too fast/slow. Evidently, these strategies work; however, it is conceivable that the training would be more efficient with more precise feedback. Despite being a complex sensorimotor task, skillful juggling relies heavily on a few measurable parameters, making it a useful model for studying motor learning in humans. 

Using VR to study and train motor skills is not new. Some studies suggests that subjects can learn simple motor tasks in Virtual Reality, at a rate comparable to that in a conventional training paradigm [14,15]. Moreover, skills acquired in VR leads to improved performance when switching to a conventional paradigm, although the performance following training in VR is slightly poorer compared to the performance when someone was trained in a conventional paradigm from the onset [15]. A number of studies shows that training within VR can improve performance and also help restore or improve motor ability in patients [16]. Nevertheless, some studies suggest that improvements in a VR context does not always generalize to real-life settings. For example, one recent study showed that playing a dart game available on HTC Vive resulted in worsening of real-life performance compared to real-life training; indeed, the dart throws actually got less accurate than dart throws prior to training in VR [17]. In other words, when it comes to the efficacy of improving motor skills through training in VR, results have been mixed. A recent review concluded that more research is necessary to study motor learning in different VR systems as well as to establish how learning using these systems translates to the real world [18].

Prior studies have examined different juggling variants in VR. In one study, the subjects learned to bounce a virtual ball on a monitor by moving a physical handle [19]. Another study, in which subjects were trained on a timing coincidence task, again showed that learning did occur and that learning in VR led to improvements outside VR [20]. Studies using other VR devices show that feedback can have a positive effect on motor learning [21], and that the VR experience feels realistic [22]. The purpose here was to see if new commercially available HMDs can be used to improve the performance on specific parameters that are key to juggling performance. Since the HTC Vive can simulate several sensory modalities, it can induce a sense of immersion—a feeling of being present in the digital world. This immersion, combined with precise motion tracking [7,8] and control over the visual and auditory feedback, makes the HTC Vive and similar HMDs promising candidates for examining what type of feedback is optimal for learning. 

In this study, we provided participants with timing and height performance feedback, presented visually to the participants within the VR environment, to test if such feedback can help participants improve their throwing technique when juggling. The control group performed the exact same task but did not receive feedback following the throws. The study was designed to test whether it is possible to learn to juggle in an HMD VR setting and whether providing feedback on parameters crucial to juggling—the timing and height of throws—can improve performance on these parameters.

## 2. Materials and Methods

### 2.1. Participants

Participants were recruited through a website dedicated to finding study participants (studentkaninen.se). People with prior juggling skills—defined as being able to juggle for more than a few catches—were not admitted to this study. This was done to ensure that existing skills did not confound the results. A total of 33 subjects participated. Of these 33 participants, 8 were excluded because they did not appear to understand the task or never grew accustomed to the VR environment. After the experiment, the participants were asked to fill out the NASA Task Load Index (TLX) [23], with one additional question added: asking how realistic the experience felt. 

### 2.2. Materials

In this study, we used the HTC Vive (HTC Corporation, Taipei, Taiwan) VR HMD system and associated controllers. The HTC Vive (Figure 1A) is a consumer VR headset with accompanying controllers and a tracking system. Though not perfect, the tracking system associated with this headset is better compared to other commercially available consumer tracking systems [7]. The head mounted display on the HTC Vive features a 100 degrees horizontal field of view, a 110 degrees vertical view, and two 1080 pixel × 1200 pixel screens. The headset was driven by a PC with an Intel i5-8400 CPU, 16 GB of RAM, and one Nvidia gtx 1070i graphics card. The simulation was able to run in a framerate of at least 90 frames per second (which is the nominal framerate using SteamVR). No framerate drops below 90 Hz were recorded). Hence the latency induced by the VR-system was expected to be around 11 ms. The HTC Vive has two controllers (Figure 1B) with multiple methods of input, such as a touchpad, a “trigger button” (shaped like the trigger of a pistol), and a “grip button” on the side of the controller (see Figure 1B). The controllers and headset are tracked by the Lighthouse tracking system, which uses two base stations that send out infrared signals that the controllers receive and convert into position data. This lets the computer track the position and rotation of the headset and controllers within the n where the base stations are located. Recent tests show that the HTC Vive is associated with accurate motion tracking [8,24,25].

### 2.3. Creating the Virtual Reality Environment

The juggling simulator was created in the game engine and development environment Unity. For certain features we used the SteamVR Unity Plugin version 2018.3.7f1 (Valve Corporation, Bellevue, WA, USA), IDE: Visual Studio 2017 (Microsoft Corporation, Redmond, WA, USA). This plugin contains several convenient GameObjects, components, and scripts (Figure 1C). For example, it contains a Player GameObject that adds a camera GameObject, whose position and rotation is automatically linked to the head mounted display. The Player GameObject also contains two hands GameObjects (Figure 1F) that are connected to the VR controllers. For the purposes of our application, we changed the model to just have the hands in red gloves without the virtual controllers, so that we could have a higher perceived realism when juggling. The SteamVR Unity Plugin also includes a system that allows developers to make any object in the environment interactable. The interaction system also has a Throwable component; when added to an object it lets the user grab and throw the object when hovering over it by pressing and releasing the trigger or grip buttons on the controllers.

### 2.4. Procedure

For a graphical illustration of the training protocol, see Figure 2. Participants were split into the two group—a feedback group and a control group—in an alternating pattern. However, since we had to exclude more participants in the control group, the last four participants were all assigned to the feedback group. The feedback group (*n* = 12) received feedback on their throws; the ‘control group’ (*n* = 13) did not receive feedback. Both groups were introduced to the juggling simulation in the same way. First, they were given only one ball to get used to the physics and interaction dynamics of the simulation (Figure 2E). The participants were asked to complete five consecutive catches with one ball, alternating hands for each throw. Because many participants had very limited VR experience, the first few practice throws were done at 60% gravity, meaning that the ball fell slower than in real life. Performance on these throws in 60% gravity were not analyzed further. After five successful catches, the gravity was increased to 100%, thus giving the subject an experience more akin to that of throwing and catching a ball in real life. 

In the two-ball stage, the subjects first watched two floating hands throw two balls in an optimal fashion. Then subjects picked up one ball in each hand and proceeded to throw the balls in arcs from one hand to the other. The subjects were not given instructions about which hand to start throwing with, but most subjects started with their dominant hand. Both groups completed a total of 200 throws with two balls. The first 50 baseline trials did not involve any feedback. After the first 50 trials, participants assigned to the feedback group received feedback on the following 100 trials; on trials 51–100, they received feedback on the height of their throws; on trials 101–150, they received feedback on the timing of their throws. The feedback was presented on the HMD screens immediately following each trial. Height feedback was presented on two bars with a green and a red zone. Two crosses represented the height of the throws from the right and left hand respectively. If the cross appeared below the midline in the green zone, it meant that the altitude was too low. If the cross was above the midline, it meant that the throw was too high. Timing feedback was presented in a similar way. A cross on a horizontal bar with green and red zones denoted the relative timing of the throws with the left and right hand. Thus, feedback was presented immediately after each trial in a way that was easy for the participants to interpret (see Figure 1G,H). On the last 50 post-test trials (151–200), performance was again assessed without any feedback. The post-test trials were used to assess whether the feedback had an effect lasting beyond the trials with the feedback.

### 2.5. Measures

A principal aim of this study was to examine if the height and timing of juggling throws can be optimized by providing immediate performance feedback in VR. But how does one define optimal height? The optimal height and relative timing of throws varies slightly depending on the height and the personal preference of the juggler. The more objects juggled, the higher the throws also need to be [9,11]. However, jugglers handling 2–3 objects throws the ball so that they see the zenith, and they throw the next ball when the previous one has just reached the zenith [11]. Based on this, combined with observations of a professional juggler (Filip Borglund), we defined the optimal height as 25 cm above eye-height and we defined the optimal time to throw the second ball as 25 ms after the first ball reached its peak. It is important to note that even professional jugglers will not throw at the exact same height each time. However, for this experiment, the most important thing was to have a reference point, to see if our participants would adapt to feedback. Before the participants started throwing the balls themselves, optimal throws as defined above, were shown to subjects in the juggling tutorial in the VR environment (see Figure 1F). To quantitatively assess the subjects’ throws, we compared the height and relative timing of each throw to the optimal throws. This gave us three measures that we used in the statistical analysis: absolute height error for the right and the left hand, and the absolute error in the relative timing of the second throw. In addition, we recorded the number of drops before each successful attempt. A drop was defined as a collision between a ball and the floor or a table. If the first ball was dropped before the second ball was thrown, an attempt would not be registered and the height and timing would not be saved. This made it possible for a person to drop more than twice per registered attempt.

## 3. Results

### 3.1. Analysis

To test whether the juggling performance improved as a result of the VR training, we used a combination of standard t-tests and linear mixed effects models to predict the number of drops, the absolute height error of throws with the right and the left hand, and the timing error of the throws. In the linear mixed effects model, we used throw number (1–200) and group (feedback/control) as fixed effects and subject as a random effect (MATLAB code is provided as Appendix A). Linear mixed effects models are statistically robust and unlike ANOVAs they take into account individual differences [26]. To assess if feedback had an effect lasting beyond the feedback session, we compared the performance of the feedback group and the control group on the last 50 trials (post-test) when neither group received any feedback.

### 3.2. Drops

Overall, there was a clear improvement in the juggling performance over the entire session (Figure 3). On average, the number of drops decreased by 0.0045 ± 0.0021 per throw (*p* < 0.0001, CI = (0.0041–0.0049)). Thus, following 200 throws subjects will on average drop 200 × 0.0045 = 0.9 balls less before a successful attempt. This was also verified in a paired t-test comparing throws 41–50 with the last ten throws (191–200). On average, the participants dropped 0.448 ± 0.507 balls less on the last 10 trials. This difference was statistically significant: t(24) = 4.42, *p* = 0.00018, CI = (0.24–0.65), Cohen’s d = 0.89. Nevertheless, there was no statistically significant difference between the feedback and the control group −0.027 ± 0.129 (*p* = 0.833, CI = (−0.281–0.226)), indicating that the feedback did not affect the number of dropped balls.

### 3.3. Height

Using two similar mixed effects models, we looked at the absolute height error for the right and the left hand. For the right hand (see Figure 4A,B), the height error decreased by 0.48 ± 0.034 mm on average per throw (*p* < 0.0001, CI = (−0.55–−0.41 mm)). Over the 200 trials, this adds up to a 200 × 0.48 = 96 mm improvement. Left hand height errors followed a similar pattern (Figure 4C,D). The error got smaller by 0.45 ± 0.037 mm per throw (*p* < 0.0001, CI = (−0.52–−0.37 mm)), or 200 × 0.45 = 90 mm over the 200 trials. Over the 200 trials, those receiving feedback did not perform significantly better than the control group (Figure 5A,B). For the right hand the difference between the feedback group and the control group was −8.7 ± 24.9 mm (*p* = 0.73, CI = (−57–40 mm)). For the left hand the difference was 3.2 ± 25.7 mm (*p* = 0.90, CI = (−40–53 mm)). In line with this, performance on the post-test trials (151–200) did not differ between the feedback and the control group (Figure 5A–C). However, within the feedback session (trials 51–100), there was a significant interaction. Participants who received feedback improved more than participants in the control group, for both hands (*p* < 0.01). To follow up on this, we ran two t-tests in which we compared the height error in the 10 trials before the feedback with the errors in the last 10 trials of the height feedback phase. For the right hand, those receiving feedback improved by 99.6 ± 139.1 mm on average while participants in the control group actually became worse by 10.8 ± 75.2 mm on average. For the left hand, the feedback improved by 109.4 ± 96.6 mm on average while the control group improved by 5.3 ± 105.6 mm in the same period. In agreement with the linear mixed effects model these were significant, for the right hand, t(23) = 2.50, *p* = 0.02, CI = (19–202 mm), Cohen’s d = 0.99; as well as for the left hand t(23) = 2.57, *p* = 0.017, CI = (20–188 mm), Cohen’s d = 1.03 (Figure 5D,E).

### 3.4. Timing

With another mixed effects model, we modelled the timing error, again using throw and group as fixed effects and subject as a random effect. As with the height, there was a small but significant decrease in the absolute timing error over the 200 trials (see Figure 4E,F). On average, the timing error got smaller by 0.068 ± 0.015 ms per trial or 200 × 0.068 = 13.6 ms over 200 trials. This effect was statistically significant (*p* < 0.0001, CI = (−0.099–−0.0386 ms)). However, there was no differences between the feedback group and the control group; not in the post-test and not within the timing feedback session (Figure 5C,F).

## 4. Discussion

The aim of this study was to examine if participants can learn to juggle in VR and whether feedback on the height and timing of the throws can enhance juggling performance. Almost all participants improved their performance. They dropped fewer balls and the height as well as the timing of their throws improved significantly over the 200 trials, suggesting that it is indeed possible to learn to juggle in a VR environment. Feedback on the height of the throws resulted in temporary enhancements. However, the relative timing of the throws did not improve as a result of timing feedback. At the end of the session, all participants dropped fewer balls, but the height and the timing of the throws did not differ between the feedback and the control group. We found some evidence that feedback in VR transiently improves juggling performance. Participants receiving height feedback did improve the height of their throws more than the control group. However, timing feedback was not associated with similar benefits, and on the post-test throws there were no differences between the performance of the feedback group and control group. The fact that the height feedback had a temporary effect suggests that it can be useful, and perhaps repeated feedback sessions could consolidate the improvements.

Some participants appeared to have difficulties focusing on the height and timing of the throws while also catching the balls. It is unclear whether these difficulties were due to the limited time given to adapt to the VR environment, or if they were due to their limited juggling experience. Again, a potential solution might be to have multiple sessions or longer sessions. VR gives researchers the opportunity to control much of the sensory input received by the subject. However, in some respects it differs from real-life practice. One major limitation of VR, especially for tasks such as juggling, is the limited peripheral vision. In our juggling exercise, participants could sometimes not see if they had actually caught the balls in their hands. The limited haptic stimulation ability of the HTC Vive controllers did not improve this matter. Recently released glove controllers from Valve allows one to track every finger separately. These improvements will add to the perceived realism of the task. Still, most participants felt that our juggling simulation—despite its limitations—was realistic. 

During the VR juggling session, participants improved on all measured parameters: the number of drops; the height of the throws; and the relative timing of the throws. In short, the participants learned the task. While some participants were unaccustomed to the VR environment and therefore needed more time to get used to it, no one experienced any motion sickness. The entire session lasted 30–40 min, and none of the participants reported experiencing any postural or visual fatigue. 

The fact that participants can learn a complex motor task in a short amount of time (~30 min) in a VR environment opens up for many possible applications. Basic motor programs for all kinds of tasks, such as playing sports, steering vehicles, or rehabilitation following injuries or strokes, might benefit from training in an immersive VR Environment [27]. Not only does training in VR allow one to provide subjects with real-time quantitative feedback; it also allows trainers or care givers to get accurate and quantitative measurements on the progress that the subject makes. Indeed, HMDs could potentially be used to assess motor performance in clinical settings; for example, when assessing people for neuropsychiatric disorders. Currently, such motor assessments are carried out by clinicians, which means that personal biases and preconceptions can affect the results. Moreover, the ability to change fundamental constants, such as gravity, or to change the sensory inputs in a controlled fashion, enables researchers to assess subjects’ ability to adapt to such changes, which can be useful for research purposes as well as training purposes. Within VR, one can directly manipulate the visual input received by the subject. This means that one could remove motor noise that has been shown to slow down motor learning [28]. This could prove particularly valuable for cerebellar patients who appear to fail to learn tasks because of increased motor noise [29]. Earlier studies have shown that training in a VR environment can be even better than training in real life, though there is also a risk of becoming over-dependent on the VR feedback [21]. Overall, the literature does not provide a clear answer as to whether performance improvements in VR successfully translates to better performance in real life [17,18]. 

## 5. Conclusions

Following a ~30 min training session in an immersive VR environment, participants exhibited clear signs of learning a simple ball juggling task. Moreover, feedback on specific performance parameters—specifically the height and timing of throws—resulted in transient improvements. These findings highlight the potential of VR technologies to improve sports performance and for rehabilitation of motor deficits. Future studies should examine if further training sessions consolidates improvements and to what extent improvements in VR translates to improvements in the real world. 

## Figures and Tables

**Figure 1 sensors-21-02966-f001:**
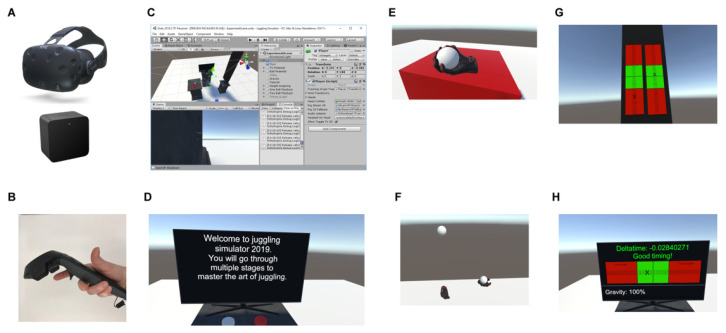
Hardware and user interface. The HTC Vive headset and the base station is shown in (**A**); the HTC hand controller is shown in (**B**). In (**C**), the interface of Unity, which was used to create the juggling simulation, is shown. (**D**) shows the TV within the VR environment that subjects received instructions on. (**E**) shows the hand and the ball used in the juggling simulator. (**F**) shows a picture of the two-ball juggling tutorial that subjects were shown in VR and asked to imitate. The height and timing feedback given to subjects in the feedback group is shown in (**G**,**H**), respectively.

**Figure 2 sensors-21-02966-f002:**
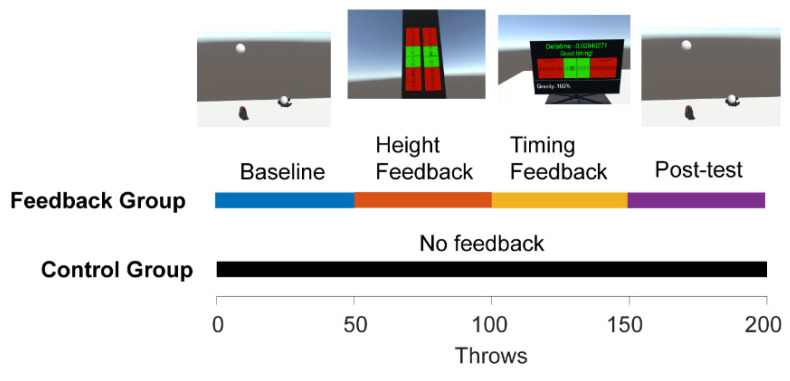
Training protocol. All participants first completed an initial practice phase in which participants got accustomed to the VR environment and learned to throw and catch one ball. Next, all participants completed 50 throws with two balls. After these 50 throws, the feedback group received height feedback on the next 50 trials and then timing feedback on the next 50 trials. The control group simply continued to practice without any quantitative input. The session finished with another 50 throws in which no feedback was given. See Figure 1 for larger versions of the screenshots.

**Figure 3 sensors-21-02966-f003:**
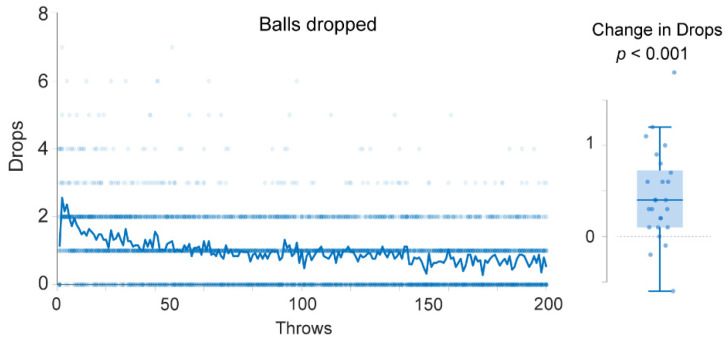
Drops. The figure shows the number of drops before each successful two-ball throw for all participants. Each dot represents the number of drops before one successful attempt for one participant. The line in the middle shows the median value. There is a clear decline in the number drops before each successful attempt as the participants spend more time in the simulator, suggesting that participants improved their performance over time. The boxplot to the right shows the distribution of change in drops. For each participant, the average number of drops on the last 10 trials (191–200) was subtracted from the number of drops on throws 41–50.

**Figure 4 sensors-21-02966-f004:**
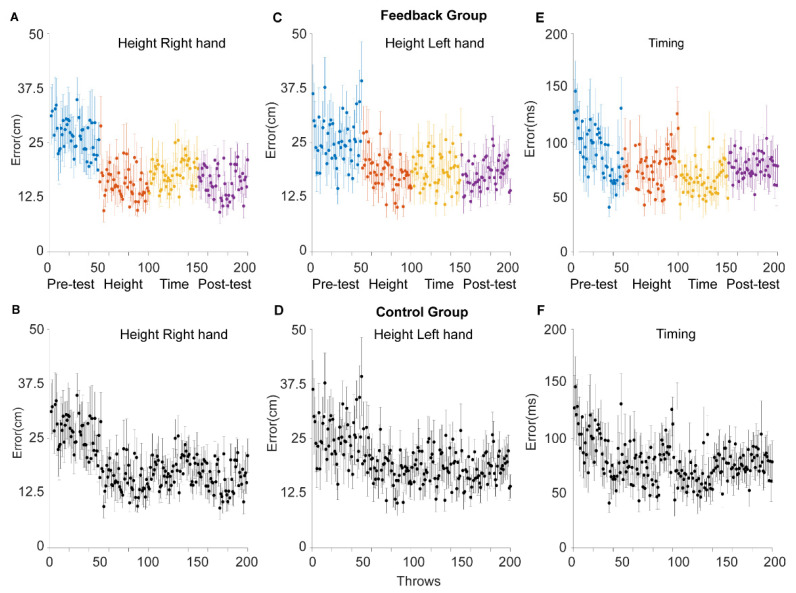
Height and timing errors (mean ± SEM) for participants receiving feedback (top row, colored lines/dots) and controls (bottom row, black lines). Panels **A**–**E** shows the difference between the optimal height of throws and the height of the actual throws with the right hand (**A**,**B**), and the left hand (**C**,**D**). Panels (**E**,**F**) shows the timing error. The color represents the stage of the experiment: blue represent baseline trials, before the participant received any feedback; red represent throws on trials where the participants received feedback on the height of their throws; yellow represent trials where participants received timing feedback; and purple represent post-feedback trials where participants did not receive any feedback.

**Figure 5 sensors-21-02966-f005:**
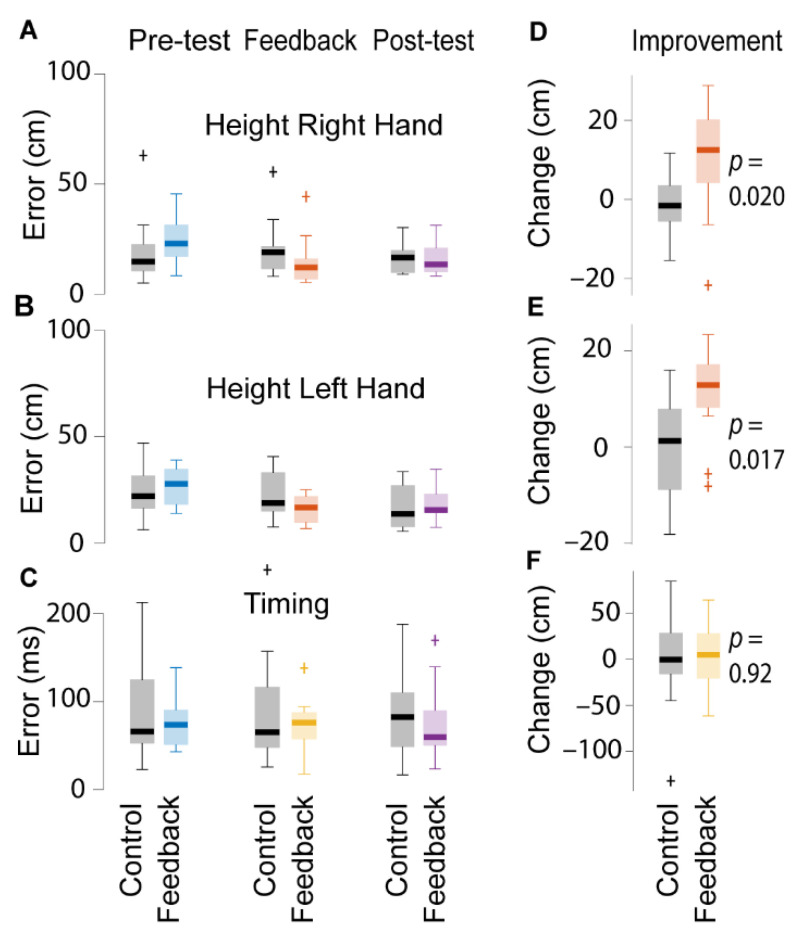
Effect of height and timing feedback. (**A**–**C**) displays the height/timing of the throws before feedback, at the end of the feedback sessions, and on the post-test. (**D**–**F**) displays how much the height/timing of the throws improved during the feedback session. This was calculated by taking the average error on the last 10 trials before the feedback session (throws 41–50) and subtracting the average error on the last 10 trials of the feedback session (41–50 for height; 91–100 for timing), and then comparing the change to the change in the control group. Participants receiving feedback on height improved the height of their throws more than those who did not receive feedback. However, this was not the case for the timing feedback.

## Data Availability

Data and code is provided as Appendix A.

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
