# Peer review of "Feedback from HTC Vive Sensors Results in Transient Performance Enhancements on a Juggling Task in Virtual Reality"

_sensors, 2021, doi:10.3390/s21092966_

Round 1

Reviewer 1 Report

A well written paper. The literature review and methodology are sound, and the results of the study are interesting. The authors are particularly commended for describing the experimental intervention in a good level of detail, which is often absent from papers. Future work should clearly include asking participants to try juggling in real life after practicing in VR. Overall, excellent work, and well worthy of being published.

Author Response

Thank you for the kind words! 

Reviewer 2 Report

This paper presents a study on learning juggling in a VR environment.

The article is well written and the study is scientifically sound.

I recommend accepting this paper after minor revisions.

The following are my comments and suggestions:

     Line 29: Please expand the acronym "EMG" here (and do that for all acronyms at their first occurrence, as you did in Line 36 for "Virtual Reality").

     Lines 36-38: Please improve the definition of VR (and provide a reference), in particular focus on distinctive characteristics of VR in respect to AR and MR (e.g., immersion).

     Lines 38-41: This statement is too strong. VR environments based on HMDs are the most common solution only in some fields (e.g., consumer VR), but not in others (e.g., CAVE systems in industrial VR).

     Line 41: Please specify more details about the device selected (e.g., specific version, manufacturer, etc.).

     Lines 65-67: This sentence is not easy to read. Please fix the punctuation and/or improve its readability.

     Line 70: Change "improves" into "improved".

     Lines 81-82: I think "systems" (or something similar) is more appropriate than "devices" here.

     Line 84: Try to be consistent with the case used for the same name: here you write "virtual reality" (sentence-case) but before you always used "Virtual Reality" (title-case). Please check entire text.

     Line 87: Change "leads" into "led".

     Lines 88-90: At the beginning of this sentence you refer to "augmented reality", whereas at the end you mention "virtual reality". Is AR a typo? If not, you have to introduce the concept of AR to the reader.

     Line 93: Here you mention "immersion" for the first time. I suggest to introduce this important VR concept before discussing about it.

     Line 97: This sentence has to be more specific about the "feedback" (e.g., visual, auditory, tactile, etc.). Also, I think that the "derived from" is misleading. 

     Line 112: Please add a reference for the TLX.

     Line 115: Figure 2 is referenced multiple times before encountering the very first reference to Figure 1; so, I suggest to swap Figure 2 and Figure 1.

     Line 121: If you mean the update-rate of the software, please refer to fps (frames-per-second) instead of Hz, to avoid confusion with the refresh rate of the displays.

     Line 124: Change reference to Figure 2 with Figure 2B.

     Lines 126-128: Typo. Please fix this sentence. Also, a picture of the base station of the tracking system should be added to Figure 2.

     Line 132: Please add more details for the Unity game engine (e.g., specific version) and the IDE used (e.g., Visual Studio, Monodevelop, etc.).

     Line 174: Please increase the size of Figure 1 so that the small screenshots are big enough, or reference Figure 2 in this caption (so the reader can jump to the high-resolution version of these pictures).

     Line 152-153: 60% gravity usually means using 60% of the standard gravity force. If so, objects move slower going down (smaller acceleration due to 60% gravity force), but faster going up (smaller deceleration due to 60% gravity force). Please correct this statement.

     Line 159: It seems there is a double-space after the period. Please remove it.

     Line 164: Please change "Heigh" into "Height".

     Line 189-190: It seems to me that these observations depend on some settings, such as: gravity, and object weight (to say the least). Please expand this sentence, especially considering the gravity tuning performed in your VR application.

     Line 257: Please use a non-breakable hyphens to avoid the hyphenation of "t-tests".

     Some considerations on the latency of the VR/HMD/tracking affecting the timing feedback should be added.

     Some considerations on the fatigue (e.g., visual, posture, etc.) induced by using VR/HMD during a training session should be added.

Reviewer 3 Report

This is an interesting, if limited study.  As such it will be of interest to readers who may well have ideas for other similar studies, that will help build a useful corpus of knowledge.  The study is presumably based on the idea that VR systems can help develop physical skills which would have wide positive benefits in sports science and rehabilitation, and it is indeed already being applied successfully, as some of the references attest.  It would probably be more useful to readers if this was emphasised in the conclusions and suggestions were made for further studies.

I found the diagrams so small that they were almost impossible to read, and would therefore suggest that they were separated and enlarged.
